# Genome-Wide Association Study of Body Conformation Traits in a Three-Way Crossbred Commercial Pig Population

**DOI:** 10.3390/ani13152414

**Published:** 2023-07-26

**Authors:** Shaoxiong Deng, Yibin Qiu, Zhanwei Zhuang, Jie Wu, Xuehua Li, Donglin Ruan, Cineng Xu, Enqing Zheng, Ming Yang, Gengyuan Cai, Jie Yang, Zhenfang Wu, Sixiu Huang

**Affiliations:** 1College of Animal Science and National Engineering Research Center for Breeding Swine Industry, South China Agricultural University, Guangzhou 510642, China; 13970439901@163.com (S.D.); 13422157044qyb@gmail.com (Y.Q.); zwzhuang@outlook.com (Z.Z.); wujiezi163@163.com (J.W.); xuehuali1998@163.com (X.L.); ruandl@stu.scau.edu.cn (D.R.); cnxu@stu.scau.edu.cn (C.X.); eqzheng@scau.edu.cn (E.Z.); cgy0415@163.com (G.C.); jieyang2012@hotmail.com (J.Y.); 2Guangdong Provincial Key Laboratory of Agro-Animal Genomics and Molecular Breeding, South China Agricultural University, Guangzhou 510642, China; 3College of Animal Science and Technology, Zhongkai University of Agriculture and Engineering, Guangzhou 510225, China; yangming@zhku.edu.cn; 4Yunfu Subcenter of Guangdong Laboratory for Lingnan Modern Agriculture, Yunfu 527400, China

**Keywords:** GWAS, MLM, FarmCPU, body conformation traits

## Abstract

**Simple Summary:**

In this study, a genome-wide association study (GWAS) was conducted on 1518 Duroc × (Landrace × Yorkshire) commercial pigs to investigate the genetic basis of body conformation traits. The traits analyzed included body length, body height, chest circumference, abdominal circumference, and waist circumference. The researchers used two statistical models, a mixed linear model (MLM) and a fixed and random model circulating probability unification (FarmCPU), to identify significant genetic variants associated with these traits. A total of 60 significant single nucleotide polymorphisms (SNPs) were discovered in the crossbred pigs. Furthermore, the researchers identified a novel significant quantitative trait locus (QTL) on chromosome SSC7 (*Sus scrofa* chromosome 7) that was specifically associated with waist circumference. These findings contribute to our understanding of the genetic mechanisms underlying body conformation traits in crossbred commercial pigs.

**Abstract:**

Body conformation is the most direct production index, which can fully reflect pig growth status and is closely related to critical economic traits. In this study, we conducted a genome-wide association study (GWAS) on body conformation traits in a population of 1518 Duroc × (Landrace × Yorkshire) commercial pigs. These traits included body length (BL), body height (BH), chest circumference (CC), abdominal circumference (AC), and waist circumference (WC). Both the mixed linear model (MLM) and fixed and random model circulating probability unification (FarmCPU) approaches were employed for the analysis. Our findings revealed 60 significant single nucleotide polymorphisms (SNPs) associated with these body conformation traits in the crossbred pig population. Specifically, sixteen SNPs were significantly associated with BL, three SNPs with BH, thirteen SNPs with CC, twelve SNPs with AC, and sixteen SNPs with WC. Moreover, we identified several promising candidate genes located within the genomic regions associated with body conformation traits. These candidate genes include *INTS10*, *KIRREL3*, *SOX21*, *BMP2*, *MAP4K3*, *SOD3*, *FAM160B1*, *ATL2*, *SPRED2*, *SEC16B*, and *RASAL2*. Furthermore, our analysis revealed a novel significant quantitative trait locus (QTL) on SSC7 specifically associated with waist circumference, spanning an 84 kb interval. Overall, the identification of these significant SNPs and potential candidate genes in crossbred commercial pigs enhances our understanding of the genetic basis underlying body conformation traits. Additionally, these findings provide valuable genetic resources for pig breeding programs.

## 1. Introduction

Body conformation traits are critical indicators of swine breeding objectives that directly reflect physical size, body structure, and development [1,2,3,4,5]. These factors are closely related to their physiological function, production performance, disease resistance, and adaptability to external living conditions. Furthermore, previous studies have revealed positive genetic correlations between body-conformation-related traits and body growth traits [6,7,8].

The pig QTL database (Release 49) currently reports 35,384 QTLs associated with 716 traits, including one hundred and fifty-eight, two hundred and twenty, thirty-five, thirty-seven, and two QTLs related to body height (BH), body length (BL), chest circumference (CC), abdominal circumference (AC), and waist circumference (WC), respectively [9]. Several studies have revealed significant genes and SNPs associated with body conformation traits using GWAS. For example, Li et al. identified 714 significantly associated SNPs located at 39 regions for body traits and seven functionally related candidate genes [10]. Zhou et al. showed that *ITGA11*, *TLE3*, and *GALC* might play a role in the body conformation traits based on single and multi-trait GWASs in two Duroc pig populations [11]. Hong et al. reported that a highly significant SNP (S17_15781294) located on *Sus scrofa* chromosome 7 (SSC7) explained 9.09% of the genetic variance for body length and 9.57% of the genetic variance for body height in Large White pigs [4]. These findings have provided multiple molecular markers to porcine breeding for body conformation traits.

Based on previous research, most GWAS studies have employed mixed linear models (MLM), a commonly used method for addressing population structure and genetic relatedness when deciphering the genetic architecture of complex traits in livestock [12]. However, this model has limitations in accurately estimating marker effects as most quantitative traits are influenced by multiple loci, leading to confounding issues [13]. The fixed and random model circulating probability unification (FarmCPU) model offers an alternative solution by separating the MLM into a fixed effect model and a random effect model that uses pseudo-quantitative trait nucleotides (QTNs) [13]. This multi-locus model has been proven to be more effective in detecting candidate genes by resolving confounding issues in numerous studies in livestock and plants [13,14,15]. Recently, many researchers have combined the two approaches to reveal more trait-related SNPs and genes [16,17].

This study aimed to detect new genetic variants and identify candidate genes for body conformation traits by utilizing MLM and FarmCPU-based genome-wide association studies in 1518 crossbred commercial pigs. This approach was taken as most previous GWAS analyses of body conformation traits were limited to a single method and only focused on purebred pigs, leading to fewer candidate genes being identified and no causal mutations found. Overall, the findings suggest that genetic factors play a significant role in determining body size and shape traits in pigs, and identifying the associated genes and pathways may contribute to improving pork production and understanding obesity in humans.

## 2. Materials and Methods

### 2.1. Ethics Statement

All animals used in this study were handled following the specifications for the care and use of experimental animals established by the Ministry of Agriculture of China. The ethics committee of South China Agricultural University (Guangzhou, China) approved this study especially. The experimental animals were not anesthetized or euthanized in this study.

### 2.2. Animals and Phenotypic Data

The experimental animals were derived from a commercial crossbred population of pigs. In brief, 84 Duroc males were mated with 397 Landrace × Yorkshire females, resulting in a large cohort of offspring (757 boars and 764 sows). All pigs were maintained under consistent feeding conditions and raised at four farms operated by Wen’s Foodstuffs Group Co., Ltd. in Guangdong, China. After reaching a fattening weight, 1518 pigs born between 2018 and 2019 were processed for phenotype recording with an average body weight of 115 kg in 13 batches. All the pigs were measured on the following traits: BH (body height) was measured from shoulder to ground; BL (body length) was measured from the midpoint of the ear to the tail [4]; the CC (chest circumference), AC (abdominal circumference), and WC (waist circumference) were measured by circling the trailing edge of the scapula, the largest part of the abdomen, and the front edge region of the hind leg in the pigs [11], respectively. All five body conformation trait measurements were performed on the same flat surface and the pig was kept in a natural standing posture during the measurement.

### 2.3. Genotypes and Quality Control

The genomic DNA of each pig was extracted from ear tissue via a standard phenol/chloroform method and was diluted to 50 ng/μL [18]. The 1518 DLY pigs were genotyped using a GeneSeek Porcine 50K BeadChip (Neogen, Lincoln, NE, USA), containing 50,703 SNPs [19]. The SNP data quality control (QC) was conducted using PLINK v1.90 software [20]. Briefly, animals and SNPs with call rates of >0.90, minor allele frequency > 0.01, and *p*-value > 10^−6^ for the Hardy–Weinberg equilibrium test were included. Furthermore, all SNPs located on the sex chromosome and unmapped regions were excluded, following our previous study [19,21,22]. After QC, 1518 pigs and 28,393 SNPs were available for further analysis.

### 2.4. Pearson’s Correlation Coefficient and Estimation of Genetic Parameters

In this study, we used Pearson’s correlation coefficient to calculate the phenotypic correlation ($r_{p\;}$) between the traits [11]. Moreover, we estimated the genetic correlation and heritability using GCTA in bivariate mode. To assess the genetic correlation between two traits, we conducted a bivariate genome-based restricted maximum likelihood (GREML) analysis and calculated the genetic correlation coefficient using the following formula [11,23]:$$r_{g\;}=\frac{\sigma_{g1g2}}{\sigma_{g1}\sigma_{g2}}$$

$r_{g\;}$ is the genetic correlation coefficient between two traits, the subscripts “1” and “2” denote the two traits, $\sigma_{g1g2}$ refers to the genetic covariance, and $\sigma_{g}$ represents the square root of the genetic variance for the trait (as captured by all SNPs).

The restricted maximum likelihood method was used to estimate the phenotypic variance explained by the significant SNPs for BL, BH, CC, AC, and WC traits using GCTA software (version 1.93.2 beta) [24]. The SNP-based heritability and phenotypic variance explained by the significant SNPs was calculated in the following model [17]:$$y=X\beta +g+\varepsilon \;\mathrm{with}\;var\left(y\right)=A_{g}\sigma_{g}^{2}+I\sigma_{\varepsilon }^{2}$$
where y refers to the vector of phenotypic values; β is a vector of fixed effects; X is an incidence matrix for β; g represents the vector of the aggregate effect of all the qualified SNPs for the pigs; 𝜀 is a vector of residual effects with$\;\varepsilon \sim N\;\left(0,\;I\sigma_{\varepsilon }^{2}\right)$; I is the identity matrix; $var\left(y\right)$ is the phenotypic variance explained by the significant SNPs or heritability; $A_{g}$ is the genetic relationship matrix (GRM); $\sigma_{g}^{2}$ corresponds to the additive genetic variance captured by either the selected SNPs or genome-wide SNPs; and $\sigma_{\varepsilon }^{2}$ refers to the residual variance.

### 2.5. Population Structure Analysis

Population stratification is a major confounding factor that can affect the reliability of GWAS. To account for this, we performed a principal component analysis (PCA) using the qualified SNPs to investigate the population structure of DLY pigs and added them as fixed effects in our analysis. A Q–Q plot was generated using the R software to assess the population stratification level.

### 2.6. Association Analyses

#### 2.6.1. MLM-Based GWAS

In the present study, the MLM was performed by using GEMMA software (version 0.98.5) [25] for five body conformation traits with the command “-lmm 1”. The statistical model is described as follows:y=Wα+Xβ+u+ε
where y is the vector of phenotypic values in DLY populations; W is the incidence matrix of covariates (fixed effects), including sex, farms, body weight at the time of measurement for the five traits, and the top-five PCAs [19]; α is a vector of the corresponding coefficients including the intercept; X is the vector of all marker genotypes; β represents the corresponding effect of marker size; u refers to an n×1 vector of random effects, with $u\sim MVNn(0,K\sigma_{g}^{2}$); and ε is an vector of errors, with $\varepsilon \sim MVNn\left(0,I\sigma_{e}^{2}\right)$. K is a genomic relatedness matrix; $\sigma_{g}^{2}$ is the additive genetic variance; I is the identity matrix; $I\sigma_{e}^{2}$ is the residual variance; n refers to the number of analyzed DLY pigs; and MVN denotes multivariate normal distribution.

#### 2.6.2. FarmCPU-Based GWAS

The GAPIT (version 3.0) R package [26,27] was used to conduct FarmCPU-based GWAS. All parameters were set as default. Briefly, the FarmCPU model consists of two parts: the fixed-effect model (FEM) and the random-effect model (REM), which is evaluated iteratively. The effects in the FEM include the top five principal components, sex, and pseudo-QTNs as follows [13,28]:$$y=Pb_{p}+M_{t}b_{t}+s_{j}d_{j}+e$$
where y is a vector of phenotypes of the analyzed trait; $b_{p}$ is a vector of fixed effects including the top-five PCAs, sex, farms, and body weight; $b_{t}$ is a vector of the fixed effects for the pseudo-QTNs; P and $M_{t}$ are the corresponding incidence matrices for $b_{p}$ and $b_{t}$, respectively; $d_{j}$ is the effect of the *j*-th candidate SNP; $s_{j}$ is the genotype for the *j*-th candidate SNP; and e is a vector of the residuals. The REM model updates the pseudo-QTNs using the SUPER (settlement of MLM under progressively exclusive relationship) algorithm as follows:y=u+e
where y is a vector of phenotypes, $u\sim MVN\left(0,2K\sigma_{u}^{2}\right)$ with $\sigma_{u}^{2}$ being the unknown genetic variance and K being the kinship matrix computed by the pseudo-QTNs, and e is a vector of the residuals.

### 2.7. Identification of Significant Single Nucleotide Polymorphisms Associated with Body Conformation Traits

To reduce the number of false negative results, the false discovery rate (FDR) was used to determine the threshold [29,30]. FDR was set as 0.01, and the threshold *p*-value was defined as P=FDR×N/M, where N is the number of SNPs with *p* < 0.01 in the GWAS results, and M refers to the total number of qualified SNPs of crossbred pigs.

### 2.8. Haplotype Block Analysis

The software Plink v1.90 [20] and Haploview4.2 [31] were used for haplotype block analysis. Linkage disequilibrium (LD) blocks were defined using Haploview4.2 based on default parameters according to the criteria of Gabriel et al. [32].

### 2.9. Candidate Gene Search and Functional Annotation

According to our previous findings, the LD in DLY pigs is typically low, with an R-squared value of 0.2 and a genetic distance of around 200 kb [19]. However, this distance is not enough to find a candidate gene. We retrieved genes within 0.5 Mb on either side of the significant SNPs using the gene annotation information of pig reference genome *Sus scrofa* 11.1 from the Ensemble genome database, accessed in October 2022 [33]. To gain insight into potential candidate genes, we performed further Gene Ontology (GO) term annotation, and Kyoto Encyclopedia of Genes and Genomes (KEGG) pathway analysis in KOBAS v3.0 [34]. Enriched terms with corrected *p*-values < 0.05, as determined by Fisher’s exact test and Benjamini–Hochberg correction, were selected for further exploration of genes involved in biological pathways and processes.

## 3. Results and Discussion

### 3.1. Phenotype Statistics and Correlations among the Traits

Table 1 presents the summary statistics of the phenotypic traits. The DLY population showed an average body length (BL) of 123.68 cm, body height (BH) of 64.61 cm, chest circumference (CC) of 112.82 cm, abdominal circumference (AC) of 121.32 cm, and waist circumference (WC) of 110.43 cm. As shown in Appendix A, our normality test results demonstrated that all phenotypic values were normally distributed. The DLY population exhibited large phenotypic diversity, with body length ranging from 101 to 145 cm and chest circumference ranging from 88 to 140 cm (Table 1). The coefficients of variation for BL, BH, CC, AC, and WC were 5.78%, 5.64%, 7.26%, 7.07%, and 7.95%, respectively (Table 1). The heritability of BL, BH, CC, AC, and WC ranged from 0.21 to 0.35, indicating moderate heritability and potential for genetic improvement. Among these traits, BL, BH, and CC showed higher heritability compared to Chinese native pigs and Large White pigs [4,5]. Moreover, the higher heritability of CC, AC, and WC compared to Duroc pigs suggested that breed-specific heritability variations may exist [11]. Figure 1 shows the genetic and phenotypic correlation coefficients among the traits. The results showed a high phenotypic and genetic correlation of CC, AC, and WC ($r_{g}>0.90,\;r_{p}>0.90$). This correlation was higher than that observed in our previous study on Duroc pigs ($r_{g}>0.77$) [11], suggesting that these traits can be simultaneously improved in pig breeding. Interestingly, BH was negatively genetically correlated with CC ($r_{g}=-0.49$), WC ($r_{g}=-0.63$), and AC ($r_{g}=-0.57$). In breeding, negative genetic correlation between traits means that selection for improvement in one trait is likely to result in a decrease in the other trait. This trade-off can be managed by setting breeding goals that prioritize which traits are most important for the specific breeding objective, and using selection indices that balance the relative weights of the traits in the breeding program. Body height and the three circumference traits show a negative genetic correlation, which means that selecting individuals with higher body height for breeding tends to result in individuals with smaller circumference traits being selected. This may be because individuals that grow faster during development tend to allocate more growth resources, resulting in slower growth in other areas [35,36].

### 3.2. Genome-Wide Association Studies for Body Conformation Traits

In our results, the Q–Q plot with genomic inflation factors (λ) showed no systematic inflation of test statistics for both methods of GWAS (Appendix A). A total of 60 SNPs surpassed the FDR threshold of 0.01 in the MLM and FarmCPU-based GWAS methods. Manhattan plots were generated to visualize the GWAS results for the five conformation traits (Figure 2A–J). Of these SNPs, sixteen were significantly associated with BL, three with BH, thirteen with CC, twelve with AC, and sixteen with WC. The MLM-based GWAS detected 20 significant SNPs, the FarmCPU-based GWAS detected 52 significant SNPs, and both methods identified 12. FarmCPU-based GWAS detected additional SNPs and confirmed most of the significant SNPs in the MLM-based GWAS, suggesting it can increase statistical power and complement MLM-based GWAS results [16]. In this study, 278 functional genes were located within 500 kb of the significant SNPs. Furthermore, we utilized the 278 functional genes to perform gene function enrichment analysis in order to identify pathways and biological processes that are associated with the five conformation traits in pigs. As a result, 11 genes were selected as promising candidate genes for body conformation traits after querying the literature for information about the association between all candidate genes’ nearest peak SNPs and the analyzed body conformation traits. These candidate genes warrant further investigation to understand the genetic architecture of these traits better.

Body length and height are typically not prioritized as target breeding traits for pigs; nevertheless, these morphological features may significantly impact the value of a sow during purchase [4]. We detected sixteen SNPs significantly associated with body length, three by MLM-based GWAS and fifteen by FarmCPU-based GWAS (Figure 2A,B and Table 2). Gene set enrichment analysis revealed many terms that might be relevant to body length, including regulation of the mitotic cell cycle, dopaminergic synapse, and other related pathways (Figure 3A and Appendix A). These pathways play important roles in cell division and differentiation, which are essential for growth and development [37]. Three SNPs (MARC0030380, ALGA0105578, and MARC0052457), located near *INTS10*, *KIRREL3*, and *SOX21*, were identified by both GWAS methods. The most significant SNP was MARC0052457 (*p* = 5.75 × 10^−8^) on SSC11 upstream of *SOX21*, explaining 1.3% of the phenotypic variance for body length. Moreover, individuals with genotype GG had a 2.51 cm increase in body length compared to individuals with genotype AA (Figure 4C). Recently, Wang et al. identified *SOX21* as a candidate gene for growth-related traits by selection signal analysis (runs of homozygosity, EigenGWAS) in 150 Laiwu pigs [38]. Mice lacking *SOX21* have reduced growth and increased energy expenditure. Moreover, further research and investigations are needed to elucidate the role of *SOX21* and its potential impact on height regulation. *BMP2* (bone morphogenetic protein 2), another candidate gene for body length, has also been identified as a promising candidate gene for carcass length in pigs [39]. Zhou et al. and Hong et al. reported that *BMP2* plays a role in bone development and is associated with body length [4,40]. Li et al. firmed the relationship of the genotype of the GWAS lead SNP rs80965549 with the expression of the *BMP* gene by transcriptomic profile analysis of porcine cartilage tissues [41]. *BMP2* can induce chondrogenic differentiation, osteogenic differentiation, and endochondral ossification in stem cells [42]. *BMP2* also regulates early myogenesis and could inhibit proliferation or induce presumptive muscle cells to undergo apoptosis, thereby inhibiting muscle development [43]. Our results, combined with the functional studies from previous work on *BMP2*, confirm that this gene is likely to be the causal gene for body length in pigs.

Using both MLM and FarmCPU-based GWAS methods, we identified three SNPs associated with body height: DIAS0000802 on SSC 3, WU_10.2_8_18963576 on SSC 8, and ALGA0081919 on SSC 14. Nearby the significant SNPs, we identified potential candidate genes *MAP4K3*, *SOD3*, and *FAM160B1*, as shown in Figure 2C,D and Table 2. One of the SNPs (DIAS0000802) was detected by both methods. *MAP4K3* mutants display phenotypes of low nutrient availability, such as reduced growth rate, small body size, and low lipid reserve [44]. *SOD3* was considered a potential candidate gene for BMI in Yorkshire pigs [45].

Chest, waist, and abdominal circumference are highly genetically correlated traits that determine animal body size and serve as indicators of fatness and leanness. For chest circumference, 13 SNPs reached the significant threshold, and these 13 significant SNPs were found to be in close proximity to 11 protein-coding genes. The MLM-based GWAS detected seven significant SNPs, and the FarmCPU-based GWAS detected ten (Figure 2E,F and Table 2). Interestingly, four of these significant SNPs were detected by both methods, suggesting their robust association with the trait, including three essential candidate genes: *ATL2*, *SPRED2*, and *SEC16B*. KEGG pathways and GO terms were enriched for candidate genes related to calcium signaling, such as positive regulation of cytosolic calcium ion concentration and calcium-mediated signaling (Figure 3B and Appendix A). Calcium signaling plays an essential role in various physiological processes [46], including growth and development. Zhou et al. found *SPRED2* was significantly associated with chest and cannon circumference [40]. *SPRED2* was mainly expressed at the leading edges of further outgrowing structures and in folds of newly forming grooves. Therefore, *SPRED2* is likely to regulate dynamic developmental processes [47]. *SPRED2*-knocked mice exhibited reduced growth and body weight, and shorter tibia length, consistent with previous studies linking body length to circumference traits. Additionally, they showed narrower growth plates compared to wild-type mice [48]. The gene *SEC16B*, which encodes for both long *SEC16L* and short *SEC16B* proteins required to transport secretory molecules from the endoplasmic reticulum (ER) to the Golgi apparatus, is closely associated with growth and obesity [49]. Many studies have reported its strong association with obesity in humans, making it a promising candidate gene for obesity-related traits [50,51,52].

Abdominal circumference is an indicator of obesity, as it reflects excessive fat accumulation in the abdominal area, resulting in abdominal obesity. In our study, 13 significant SNPs were found distributed on nine chromosomes (Figure 2G,H and Table 2). The most significant SNP detected by FarmCPU is WU_10.2_9_131985977 (*p*-value = 2.42 × 10^−5^), which explained 0.3% of the phenotypic variance of AC. GO terms were enriched for candidate genes involved in urate metabolic processes and actin filament network formation (Figure 3C and Appendix A). The Pearson correlation between CC and AC reached 0.91, and the genetic correlation was as high as 0.96, indicating that the significant SNPs may jointly influence multiple body conformation traits. *RASAL2* is located 110 kb away from *SEC16B* on SSC 1 mentioned above. The close physical proximity of these two genes suggests that they are likely to co-regulate each other, thereby influencing abdominal circumference and chest circumference. In a combined analysis of Mexican-mestizo children and adults, *RASAL2* was significantly associated with waist circumference [53] and positively associated with body mass index in a genome-wide association study in humans [54]. *RASAL2* mutant mice showed a drastic decrease in *RASAL2* expression and a lean phenotype, displaying decreased adiposity and resistance to high-fat diet-induced metabolic disorders [55].

We identified 16 SNPs that were significantly associated with waist circumference. Only six of these SNPs were identified by the MLM-based GWAS, while eleven were detected by the FarmCPU-based GWAS (Figure 2I,J and Table 2). The most significant SNP, detected by MLM and FarmCPU, was ASGA0062816 (*p* = 2.42 × 10^−5^) on SSC9, which explained 1.63% of the phenotypic variance of WC. These significant SNPs were annotated to 12 coding genes, including three promising candidate genes: *CAB39*, *GRP*, and *ABCD4*. The enriched GO terms included activation of phospholipase C activity and threonine catabolic processes (Figure 3D and Appendix A). These pathways involve various biological processes, including lipid metabolism and energy homeostasis [56], which have been linked to obesity in humans and other animals. In humans, individuals who carry minor allele (A) of the Ca binding protein 39 (*CAB39*) rs6722579 have a higher risk of abdominal obesity (defined as waist circumference >90 cm and 80 cm in males and females, respectively) than those who do not carry the SNP [57]. *GRP* stimulates the release of gastrin and other gastrointestinal hormones, which affect food intake and may lead to anorexia, bulimia, and obesity if the gene is lacking [58]. In this study, four SNPs associated with waist circumference are located in a QTL region on SSC7 between 97.57 and 97.65 Mb (Sscofa 11.1). Figure 4A,B is a region plot of this QTL and shows the LD pattern between the GWAS peak (WU_10.2_7_103232787) and other significant SNPs, together with the most promising candidate gene *ABCD4*. This region contained six SNPs, four of which were located within *ABCD4*, and the other two SNPs were located upstream or downstream of *ABCD4*. The top site of the haplotype block explained the largest variance of phenotypic variation in SNPs, reaching up to 1.8%. Moreover, individuals with genotype CC had a 2.51 cm increase in waist circumference compared to individuals who carry genotype AA (Figure 4D). Numerous studies have shown that this gene is associated with rib number and total teat number traits [4,59,60].

## 4. Conclusions

This study performed two GWAS methods for five conformation traits in crossbred commercial pigs. As a result, we identified 60 SNPs significantly associated with five body conformation traits. Furthermore, *INTS10*, *KIRREL3*, *SOX21*, *BMP2*, *MAP4K3*, *SOD3*, *FAM160B1*, *ATL2*, *SPRED2*, *SEC16B*, *RASAL2*, *CAB39*, *GRP*, and *ABCD4* might be promising candidate genes that compose the underlying genetic architecture of porcine body conformation traits. Additionally, a novel significant quantitative trait locus (QTL) for waist circumference was detected on SSC7 within an 84 kb interval. We expect these findings can help scholars understand the genetic basis of porcine body conformation traits and could be applied in pig breeding programs.

## Figures and Tables

**Figure 1 animals-13-02414-f001:**
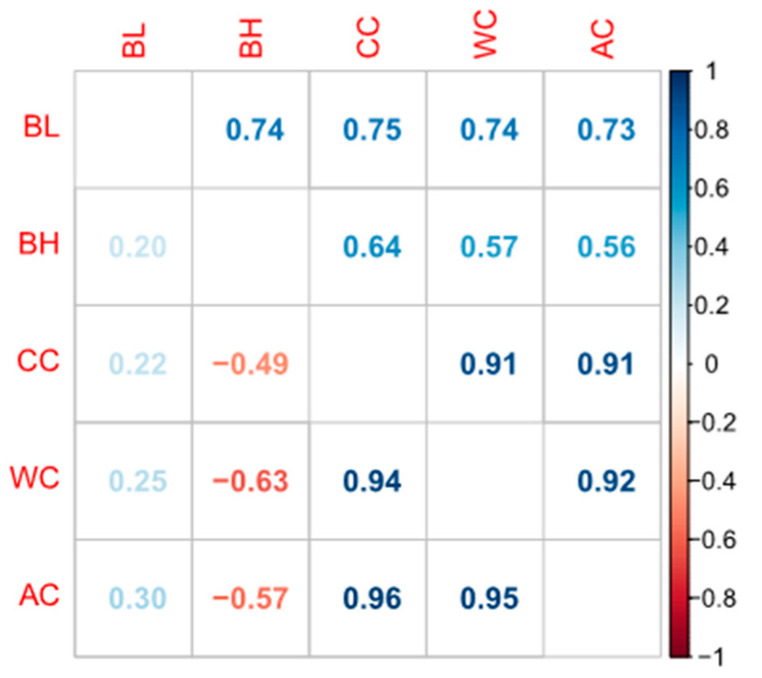
Phenotypic (above diagonal) and genetic (below diagonal) correlations between the body length, body height, chest, abdominal, and waist circumference traits in the DYL pig population.

**Figure 2 animals-13-02414-f002:**
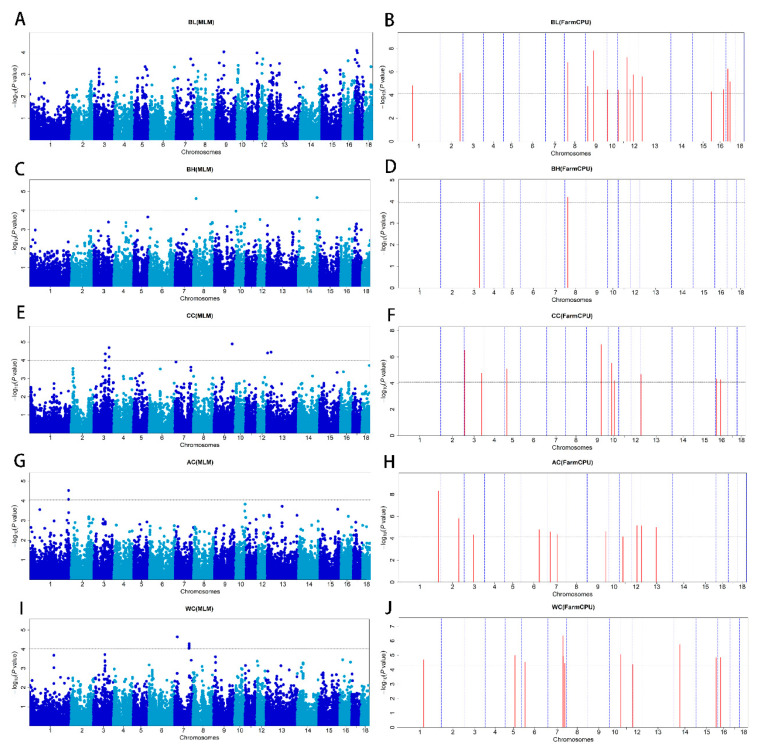
Manhattan plots illustrating the GWAS results for body length (BL), body height (BH), chest circumference (CC), abdominal circumference (AC), and waist circumference (WC) in DLY pigs, using both the MLM and FarmCPU methods. (**A**,**B**) represent the GWAS results conducted by MLM-based GWAS (threshold: *p* = 1.20 × 10^−4^) and FarmCPU-based GWAS (threshold: *p* = 7.71 × 10^−5^) for BL, respectively. (**C**,**D**) represent the GWAS results conducted by MLM-based GWAS (threshold: *p* = 9.4427 × 10^−5^) and FarmCPU-based GWAS (threshold: *p* = 1.13 × 10^−4^) for BH, respectively. (**E**,**F**) represent the GWAS results conducted by MLM-based GWAS (threshold: *p* = 8.44 × 10^−5^) and FarmCPU-based GWAS (threshold: *p* = 1.14 × 10^−4^) for CC, respectively. (**G**,**H**) represent the GWAS results conducted by MLM-based GWAS (threshold: *p* = 9.12 × 10^−5^) and FarmCPU-based GWAS (threshold: *p* = 7.08 × 10^−5^) for AC, respectively. (**I**,**J**) represent the GWAS results conducted by MLM-based GWAS (threshold: *p* = 9.55 × 10^−5^) and FarmCPU-based GWAS (threshold: *p* = 5.18 × 10^−5^) for WC, respectively. The *x*-axis represents the chromosomes, and the *y*-axis represents the $-\log10\;\left(p-\mathrm{value}\right)$.

**Figure 3 animals-13-02414-f003:**
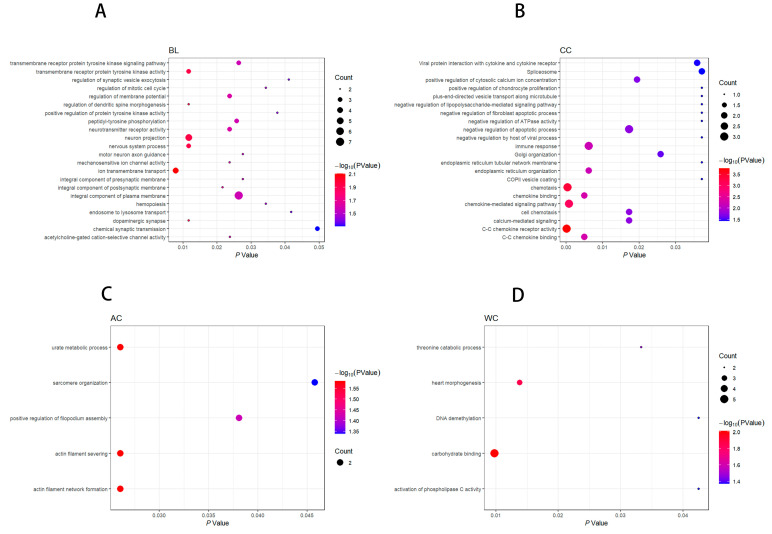
(**A**–**D**) are the bubble charts of the Gene Ontology (GO) terms and Kyoto Encyclopedia of Genes and Genomes (KEGG) pathways of the candidate genes associated with BL, CC, AC, and WC traits, respectively.

**Figure 4 animals-13-02414-f004:**
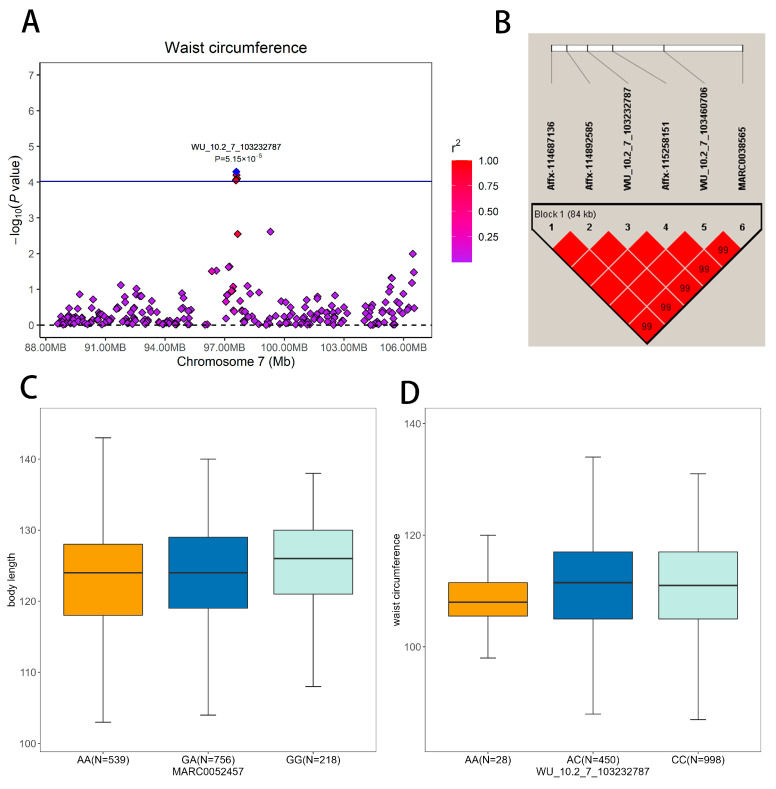
(**A**) Regional plots were constructed for the WU_10.2_7_103232787 SNP located at 88.00–106.00 Mb on SSC7 in DLY pigs. The plot displays the association signals and linkage disequilibrium (LD) between the SNP and waist circumference. (**B**) The plot indicated an 84 kb linkage disequilibrium block in the significant region on SSC7. (**C**) A boxplot was used to demonstrate the differences in body length among the three genotypes of the (MARC0052457) identified in the GWAS analysis. (**D**) A boxplot was used to demonstrate the differences in waist circumference among the three genotypes of the top SNP (WU_10.2_7_103232787).

**Table 1 animals-13-02414-t001:** Summary statistics of body conformation traits in DLY pigs.

Traits ^1^	Mean ± SD ^2^	Min ^3^	Max ^4^	CV ^5^ %	*h*² ± SE ^6^
BL	123.68 ± 7.15	101	145	5.78	0.35 ± 0.04
BH	64.61 ± 3.64	51	78	5.64	0.31 ± 0.05
CC	112.82 ± 24.24	88	140	7.26	0.34 ± 0.04
AC	121.32 ± 8.58	94	150	7.07	0.26 ± 0.04
WC	110.43 ± 8.78	84	140	7.95	0.21 ± 0.04

^1^ Five body conformation traits. ^2^ Mean ± standard deviation. ^3^ Minimum (Min). ^4^ Maximum (Max). ^5^ Coefficient of variation. ^6^ Heritability ± standard error.

**Table 2 animals-13-02414-t002:** Description of SNPs significantly associated with BL, BH, CC, AC, and WC in DLY pigs.

Trait	SSC ^1^	SNP	Location	*p*-Value	*p*-Value	R² (%) ^3^	Distance (bp)	Nearest Gene
			(bp) ^2^	(MLM)	(FarmCPU)			
BL	17	MARC0030380	12,149,145	8.19 × 10^−5^	5.77 × 10^−7^	1.91	61,261	*INTS10*
	9	ALGA0105578	54,113,499	9.50 × 10^−5^	1.54 × 10^−8^	1.42	50,019	*KIRREL3*
	17	WU_10.2_17_17479009	15,827,454	1.05 × 10^−4^		1.18	66,239	*BMP2*
	11	MARC0052457	64,090,448	1.06 × 10^−4^	5.75 × 10^−8^	1.36	263,981	*SOX21*
	8	H3GA0024522	22,446,465		1.51 × 10^−7^	0.57	NA	*NA*
	2	ALGA0118729	131,130,527		1.31 × 10^−6^	1.42	114,097	*SLC12A2*
	12	WU_10.2_12_23896898	24,026,238		1.77 × 10^−6^	1.14	33,694	*OSBPL7*
	13	WU_10.2_13_22498141	20,661,904		2.71 × 10^−6^	1.51	272,967	*ARPP21*
	17	DRGA0016669	24,960,133		7.09 × 10^−6^	1.73	276,584	*MACROD2*
	1	H3GA0002350	95,927,556		1.53 × 10^−5^	1.15	23,858	*RNF165*
	9	H3GA0026707	16,164,242		1.71 × 10^−5^	0.72	NA	*NA*
	12	WU_10.2_12_4071530	4,324,076		3.37 × 10^−5^	0.03	Within	*SEPTIN9*
	16	WU_10.2_16_67817952	62,516,694		3.47 × 10^−5^	1.16	53,119	*ATP10B*
	10	MARC0041569	5,298,516		3.78 × 10^−5^	0.92	455,315	*KCTD3*
	11	WU_10.2_11_5570350	5,881,250		3.88 × 10^−5^	1.41	72,493	*POMP*
	15	WU_10.2_15_136877153	123,439,329		5.39 × 10^−5^	1.19	63,812	*EPHA4*
BH	14	ALGA0081919	125,132,825	2.08 × 10^−5^		1.13	25,823	*FAM160B1*
	8	WU_10.2_8_18963576	18,731,675	2.36 × 10^−5^	6.32 × 10^−5^	1.88	64,961	*SOD3*
	3	DIAS0000802	101,049,861	1.08 × 10^−4^		1.93	Within	*MAP4K3*
CC	9	H3GA0028170	119,852,713	1.27 × 10^−5^	9.09 × 10^−7^	1.23	9783	*SEC16B*
	3	H3GA0010240	102,073,609	2.03 × 10^−5^		2.15	43,940	*ATL2*
	13	ALGA0119302	29,363,145	3.50 × 10^−5^	7.21 × 10^−6^	1.75	20,158	*CCR5*
	13	ASGA0055780	6,014,822	3.90 × 10^−5^	1.95 × 10^−6^	0.81	89,715	*KCNH8*
	3	MARC0004483	76,624,951	4.32 × 10^−5^		2.06	Within	*SPRED2*
	3	ASGA0015185	76,651,363	4.32 × 10^−5^		2.06	1713	*SPRED2*
	3	WU_10.2_3_108307418	102,136,805	6.36 × 10^−5^	5.43 × 10^−6^	2.07	58,936	*CYP1B1*
	10	WU_10.2_10_67005939	61,139,023		1.33 × 10^−6^	0.31	NA	*NA*
	5	INRA0019282	40,871,511		2.18 × 10^−6^	0.69	772	*SYT10*
	18	ALGA0098775	50,846,238		3.65 × 10^−6^	1.81	15,228	*CAMK2B*
	1	WU_10.2_1_179575045	161,987,727		3.74 × 10^−6^	0.7	2555	*ZNF532*
	1	ASGA0001418	16,903,857		4.07 × 10^−5^	0.2	63,327	*UST*
	2	WU_10.2_2_21124019	19,419,332		4.28 × 10^−5^	1.3	425,106	*API5*
AC	1	ALGA0009765	258,153,534	2.98 × 10^−5^		1.26	412,557	*ASTN2*
	1	WU_10.2_1_289532755	257,687,154	8.53 × 10^−5^	5.04 × 10^−9^	1.73	Within	*ASTN2*
	2	MARC0066799	115,758,230		1.59 × 10^−6^	1.71	76,874	*WDR36*
	12	MARC0115537	37,253,607		6.72 × 10^−6^	1.34	430,372	*C17orf64*
	13	ALGA0067602	5,297,429		7.42 × 10^−6^	1.77	16,929	*SATB1*
	13	MARC0021524	99,647,029		1.02 × 10^−5^	0.88	278,346	*C3orf80*
	6	MARC0000035	120,909,790		1.63 × 10^−5^	0.78	Within	*KIAA1328*
	9	WU_10.2_9_131985977	120,299,004		2.42 × 10^−5^	0.33	Within	*RASAL2*
	7	DRGA0007316	20,219,344		2.58 × 10^−5^	0.41	Within	*CARMIL1*
	7	MARC0033686	64,847,978		4.51 × 10^−5^	0.13	19,104	*SRP54*
	3	ASGA0014859	59,618,741		4.74 × 10^−5^	0.67	Within	*KCMF1*
	11	ALGA0124549	25,293,190		6.91 × 10^−5^	1.59	Within	*VWA8*
WC	7	ALGA0039140	18,645,244	2.25 × 10^−5^		1.58	362,537	*NRSN1*
	7	WU_10.2_7_103232787	97,584,287	5.15 × 10^−5^		1.84	Within	*ABCD4*
	7	Affx-115258151	97,595,573	6.36 × 10^−5^	4.58 × 10^−7^	1.63	9894	*ABCD4*
	7	WU_10.2_7_103460706	97,617,907	7.98 × 10^−5^		1.62	32,228	*ABCD4*
	7	Affx-114892585	97,575,068	8.58 × 10^−5^		1.61	Within	*ABCD4*
	7	Affx-114687136	97,568,284	8.99 × 10^−5^		1.62	Within	*ABCD4*
	14	ASGA0062816	38,090,626		1.79 × 10^−6^	1.63	Within	*RBM19*
	11	ASGA0049251	3,324,036		9.00 × 10^−6^	1.12	Within	*ATP8A2*
	5	WU_10.2_5_65149069	62,312,551		1.03 × 10^−5^	0.32	54,860	*KLRB1*
	7	WU_10.2_7_105348813	99,303,783		1.22 × 10^−5^	1.71	21,786	*GPATCH2L*
	16	ASGA0072515	19,669,219		1.42 × 10^−5^	0.82	Within	*ADAMTS12*
	15	ALGA0088031	131,517,298		1.47 × 10^−5^	0.69	24,648	*CAB39*
	1	DIAS0002061	161,757,996		2.05 × 10^−5^	2.2	36,214	*GRP*
	6	WU_10.2_6_21220801	22,691,873		3.09 × 10^−5^	1.02	231,267	*CDH8*
	7	WU_10.2_7_118076533	111,437,802		3.68 × 10^−5^	2.48	Within	*FOXN3*
	12	ALGA0064332	2,354,756		4.46 × 10^−5^	1.13	Within	*CCDC40*

Genes nearest the significant SNPs are italicized. ^1^ *Sus scrofa* chromosome. ^2^ The positions of the associated SNPs on the *Sus scrofa* Build 11.1 assembly. ^3^ Proportion of total phenotypic variation explained by each SNP.

## Data Availability

The data that support the findings of this study are available from the corresponding author upon reasonable request.

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
