# Peer review of "Genome-Wide Association Study of Body Conformation Traits in a Three-Way Crossbred Commercial Pig Population"

_animals, 2023, doi:10.3390/ani13152414_

Round 1
Reviewer 1 Report
General Comments
This is a very well written and laid out paper. It provides excellent contribution to the body of literature through the validation of previous research and the identification of a novel QTL. The references cited and relevant and recent. I thoroughly enjoyed reading this manuscript. With a few minor amendments I think this will be well received by the academic and industry communities.
One major question I have, was there a reason you focused on the 13 SNP for chest, and abdominal circumference (identified in two separate methods) as opposed to the 4 SNP that were identified in both methods? A similar question for the waist circumference. Please add some discussion related to this decision in the paper.
Specific Comments:
L35: change “we detect” to “we detected”
L97: Please re-define CC, AC and WC after the abstract. Abbreviations should be defined in the main body text in addition to the abstract.
L126: Please define “? ???ℎ ???(?)”
L135: Were the PCA results added to the fixed effects? Please clarify in your text.
L281 to 282: The sentence related to stature in humans seems out of place and not well described. Please re-write to connect these findings to the previous two sentences and your results. Please also add more discussion around the significance of your SOX21, as you did for BMP2.
L314: replace the , with a . after “13 SNPs reached the significant threshold”
L316 – 318: this sentence is a bit confusing as you are bouncing between the 4 SNP that were found by both methods and then discussing 11 protein-coding genes that are near 13 SNP (I am assuming that is from the combination of both methods?) I think this section needs to be re-worked to provide clarity.
Reviewer 2 Report
The authors presented MLM and FarmCPU-based genome-wide association studies for five body conformation traits in crossbred commercial pigs. This work provides new information to the body of work. The results of this study are interesting and provide new insight to the scientific community. However, I would like the authors to include more details regarding the following:
Line19: The abbreviation "SSC7" appears for the first time. Please write the full name.
Line74: It is unclear how these traits could be utilized to enhance pork production and their contribution to economic traits. Can you explain it?
Line91-93: The full names of BH, BL, CC, AC and WC should be provided in M&M at the first time.
Line104: Please carefully review the quality control standards as more than 20,000 SNPs were removed after quality control, which significantly impacts subsequent analysis. Check the parameters because the authors had stricter criteria for genotyping quality control.
Line110: The phrase "and calculated the genetic correlation coefficient" can be removed without affecting the meaning.
Line122:This formula functions as both a calculation method for heritability and a method for determining the proportion of phenotypic variance. The distinction lies in its application of either genome-wide data or individual SNPs to compute heritability. These two components can be integrated and discussed concurrently.
Line144: The random effect "u" differs from those represented in the formula. Please make the correction.
Line161: Please provide the full name of SUPER.
Line185: please Explain how p-values were corrected for the enriched terms.
Line189-191: The full names of BH, BL, CC, and AC should be introduced for the first time in the M&M section, while the abbreviations can be used directly in the Conclusions section.
Line347: RASAL2 is located 110 kb away from SEC16B on SSC 1, as mentioned above. This statement indicates the close proximity of these two genes and suggests the possibility of their mutual regulation. While the result is mentioned in the text, it is not further described.
Moderate editing of English language required
